# Pseudo-Labeling and Confirmation Bias in Deep Semi-Supervised Learning

## Abstract

Semi-supervised learning, i.e. jointly learning from labeled an unlabeled samples, is an active research topic due to its key role on relaxing human supervision. In the context of image classification, recent advances to learn from unlabeled samples are mainly focused on consistency regularization methods that encourage invariant predictions for different perturbations of unlabeled samples. We, conversely, propose to learn from unlabeled data by generating soft pseudo-labels using the network predictions. We show that a naive pseudo-labeling overfits to incorrect pseudo-labels due to the so-called confirmation bias and demonstrate that mixup augmentation and setting a minimum number of labeled samples per mini-batch are effective regularization techniques for reducing it. The proposed approach achieves state-of-the-art results in CIFAR-10/100, SVHN, and Mini-ImageNet despite being much simpler than other methods. These results demonstrate that pseudo-labeling alone can outperform consistency regularization methods, while the opposite was supposed in previous work. Code will be made available.

## 1 Introduction

Convolutional neural networks (CNNs) have become the dominant approach in computer vision (Lin et al., 2017; Liu et al., 2018; Kim et al., 2018; Xie et al., 2018). To best exploit them, vast amounts of labeled data are required. Obtaining such labels, however, is not trivial, and the research community is exploring alternatives to alleviate this (Li et al., 2017; Oliver et al., 2018; Liu et al., 2019).

Knowledge transfer via deep domain adaptation (Wang & Deng, 2018) is a popular alternative that seeks to learn transferable representations from source to target domains by embedding domain adaptation in the learning pipeline. Other approaches focus exclusively on learning useful representations from scratch in a target domain when annotation constraints are relaxed (Oliver et al., 2018; Arazo et al., 2019; Gidaris et al., 2018). Semi-supervised learning (SSL) (Oliver et al., 2018) focuses on scenarios with sparsely labeled data and extensive amounts of unlabeled data; learning with label noise (Arazo et al., 2019) seeks robust learning when labels are obtained automatically and may not represent the image content; and self-supervised learning (Gidaris et al., 2018) uses data supervision to learn from unlabeled data in a supervised manner. This paper focuses on SSL for image classification, a recently very active research area (Li et al., 2019).

SSL is a transversal task for different domains including images (Oliver et al., 2018), audio (Zhang et al., 2016), time series (González et al., 2018), and text (Miyato et al., 2016). Recent approaches in image classification primarily focus on exploiting the consistency in the predictions for the same sample under different perturbations (consistency regularization) (Sajjadi et al., 2016; Li et al., 2019), while other approaches directly generate labels for the unlabeled data to guide the learning process (pseudo-labeling) (Lee, 2013; Iscen et al., 2019). These two alternatives differ importantly in the mechanism they use to exploit unlabeled samples. Consistency regularization and pseudo-labeling approaches apply different strategies such as a warm-up phase using labeled data (Tarvainen & Valpola, 2017; Iscen et al., 2019), uncertainty weighting (Shi et al., 2018; Li et al., 2019), adversarial attacks (Miyato et al., 2018; Qiao et al., 2018), or graph-consistency (Luo et al., 2018; Iscen et al., 2019). These strategies deal with confirmation bias (Tarvainen & Valpola, 2017; Li et al., 2019), also known as noise accumulation (Zhang et al., 2016). This bias stems from using incorrect predictions on unlabeled data for training in subsequent epochs and, thereby increasing confidence in incorrect predictions and producing a model that will tend to resist new changes.

This paper explores pseudo-labeling for semi-supervised deep learning from the network predictions and shows that, contrary to previous attempts on pseudo-labeling (Iscen et al., 2019; Oliver et al., 2018; Shi et al., 2018), simple modifications to prevent confirmation bias lead to state-of-the-art performance without adding consistency regularization strategies. We adapt the approach proposed by Tanaka et al. (2018) in the context of label noise and apply it exclusively on unlabeled samples. Experiments show that this naive pseudo-labeling is limited by confirmation bias as prediction errors are fit by the network. To deal with this issue, we propose to use mixup augmentation (Zhang et al., 2018) as an effective regularization that helps calibrate deep neural networks (Thulasidasan et al., 2019) and, therefore, alleviates confirmation bias. We find that mixup alone does not guarantee robustness against confirmation bias when reducing the amount of labeled samples or using certain network architectures (see Subsection 4.3), and show that, when properly introduced, dropout regularization (Srivastava et al., 2014) and data augmentation mitigates this issue. Our purely pseudo-labeling approach achieves state-of-the-art results (see Subsection 4.4) without requiring multiple networks (Tarvainen & Valpola, 2017; Qiao et al., 2018; Li et al., 2019; Verma et al., 2019), nor does it require over a thousand epochs of training to achieve peak performance in every dataset (Athiwaratkun et al., 2019; Berthelot et al., 2019), nor needs many (ten) forward passes for each sample (Li et al., 2019). Compared to other pseudo-labeling approaches, the proposed approach is simpler in that it does not require graph construction and diffusion (Iscen et al., 2019) or combination with consistency regularization methods (Shi et al., 2018), but still achieves state-of-the-art results.

## 2 RELATED WORK

This section reviews closely related SSL methods, i.e. those using deep learning with mini-batch optimization over large image collections. Previous work on deep SSL differ in whether they use consistency regularization or pseudo-labeling to learn from the unlabeled set (Iscen et al., 2019), while they all share the use of a cross-entropy loss (or similar) on labeled data.

**Consistency regularization**  Imposes that the same sample under different perturbations must produce the same output. This idea was used in (Sajjadi et al., 2016) where they apply randomized data augmentation, dropout, and random max-pooling while forcing softmax predictions to be similar. A similar idea is applied in (Laine & Aila, 2017), which also extends the perturbation to different epochs, i.e. the current prediction for a sample has to be similar to an ensemble of predictions of the same sample in the past. Here the different perturbations come from networks at different states, dropout, and data augmentation. In (Tarvainen & Valpola, 2017), the temporal ensembling method is interpreted as a teacher-student problem where the network is both a teacher that produces targets for the unlabeled data as a temporal ensemble, and a student that learns the generated targets by imposing the consistency regularization. Tarvainen & Valpola (2017) naturally re-define the problem to deal with confirmation bias by separating the teacher and the student. The teacher is defined as a different network with similar architecture whose parameters are updated as an exponential moving average of the student network weights. This method is extended in (Li et al., 2019), where they apply an uncertainty weight over the unlabeled samples to learn from the unlabeled samples with low uncertainty (i.e. entropy of the predictions for each sample under random perturbations). Additionally, Miyato et al. (2018) use virtual adversarial training to carefully introduce perturbations to data samples as adversarial noise and later impose consistency regularization on the predictions. More recently, Luo et al. (2018) propose to use a contrastive loss on the predictions as a regularization that forces predictions to be similar (different) when they are from the same (different) class. This method extends the consistency regularization previously considered only in-between the same data samples to in-between different samples. Their method can naturally be combined with (Tarvainen & Valpola, 2017) or (Miyato et al., 2018) to boost their performance. Similarly, Verma et al. (2019) propose interpolation consistency training, a method inspired by (Zhang et al., 2018) that encourage predictions at interpolated unlabeled samples to be consistent with the interpolated predictions of individual samples. Also, authors in (Berthelot et al., 2019) apply consistency regularization by guessing low-entropy labels, generating data-augmented unlabeled examples and mixing labeled and unlabeled examples using mixup (Zhang et al., 2018). Both (Verma et al., 2019) and (Berthelot et al., 2019) adopt (Tarvainen & Valpola, 2017) to estimate the targets used in the consistency regularization.

Co-training (Qiao et al., 2018) uses two (or more) networks trained simultaneously to agree on their predictions (consistency regularization) and disagree on their errors. Errors are defined as

different predictions when exposed to adversarial attacks, thus forcing different networks to learn complementary representations for the same samples. Recently, Chen et al. (2018) measure the consistency between the current prediction and an additional prediction for the same sample given by an external memory module that keeps track of previous representations. They additionally introduce an uncertainty weighting of the consistency term to reduce the contribution of uncertain predictions. Consistency regularization methods such as (Laine & Aila, 2017; Tarvainen & Valpola, 2017; Miyato et al., 2018) have all been shown to benefit from stochastic weight averaging method (Athiwaratkun et al., 2019), that averages network parameters at different training epochs to move the SGD solution on borders of flat loss regions to their center and improve generalization.

**Pseudo-labeling** Seeks the generation of labels or pseudo-labels for unlabeled samples to guide the learning process. An early attempt at pseudo-labeling proposed in (Lee, 2013) uses the network predictions as labels. However, they constrain the pseudo-labeling to a fine-tuning stage, i.e. there is a pre-training or warm-up to initialize the network. A recent pseudo-labeling approach proposed in (Shi et al., 2018) uses the network class prediction as hard labels for the unlabeled samples. They also introduce an uncertainty weight for each sample loss, it being higher for samples that have distant $k$-nearest neighbors in the feature space. They further include a loss term to encourage intra-class compactness and inter-class separation, and a consistency term between samples with different perturbations. Improved results are reported in combination with (Tarvainen & Valpola, 2017). Finally, a recently published work (Iscen et al., 2019) implements pseudo-labeling through graph-based label propagation. The method alternates between two steps: training from labeled and pseudo-labeled data and using the representations of the network to build a nearest neighbor graph where label propagation is applied to refine hard pseudo-labels. They further add an uncertainty score for every sample (softmax prediction entropy based) and class (class population based) to deal, respectively, with the unequal confidence in network predictions and class-imbalance.

## 3 PSEUDO-LABELING

We formulate SSL as learning a model $h_\theta(x)$ from a set of $N$ training samples $\mathcal{D}$. These samples are split into the unlabeled set $\mathcal{D}_u = \{x_i\}_{i=1}^{N_u}$ and the labeled set $\mathcal{D}_l = \{(x_i, y_i)\}_{i=1}^{N_l}$, being $y_i \in \{0, 1\}^C$ the one-hot encoding label for $C$ classes corresponding to $x_i$ and $N = N_l + N_u$. In our case, $h_\theta$ is a CNN and $\theta$ represents the model parameters (weights and biases). As we seek to perform pseudo-labeling, we assume that a pseudo-label $\tilde{y}$ is available for the $N_u$ unlabeled samples. We can then reformulate SSL as training using $\tilde{\mathcal{D}} = \{(x_i, \tilde{y}_i)\}_{i=1}^{N}$, being $\tilde{y} = y$ for the $N_l$ labeled samples.

The CNN parameters $\theta$ can be optimized using categorical cross-entropy:

$$\ell^*(\theta) = -\sum_{i=1}^{N} \tilde{y}_i^T \log\left(h_\theta(x_i)\right), \tag{1}$$

where $h_\theta(x)$ are the softmax probabilities produced by the model and $\log(\cdot)$ is applied element-wise. A key decision is how to generate the pseudo-labels $\tilde{y}$ for the $N_u$ unlabeled samples. Previous approaches have used hard pseudo-labels (i.e. one-hot vectors) directly using the network output class (Lee, 2013; Shi et al., 2018) or the class estimated using label propagation on a nearest neighbor graph (Iscen et al., 2019). We adopt the former approach, but use soft pseudo-labels, as we have seen this outperforms hard labels, confirming the observations noted in (Tanaka et al., 2018) in the context of relabeling when learning with label noise. In particular, we store the softmax predictions $h_\theta(x_i)$ of the network in every mini-batch of an epoch and use them to modify the soft pseudo-label $\tilde{y}$ for the $N_u$ unlabeled samples at the end of every epoch. We proceed as described from the second to the last training epoch, while in the first epoch we use the softmax predictions for the unlabeled samples from a model trained in a 10 epochs warm-up phase using the labeled data subset $\mathcal{D}_u$.

We use the two regularizations applied in (Tanaka et al., 2018) to improve convergence. The first regularization deals with the difficulty of converging at early training stages when the network's predictions are mostly incorrect and the CNN tends to predict the same class to minimize the loss. Assignment of all samples to a single class is discouraged by adding:

$$R_A = \sum_{c=1}^{C} p_c \log\left(\frac{p_c}{\overline{h}_c}\right), \tag{2}$$

where $p_c$ is the prior probability distribution for class $c$ and $\overline{h}_c$ denotes the mean softmax probability of the model for class $c$ across all samples in the dataset. As in (Tanaka et al., 2018), we assume a uniform distribution $p_c = 1/C$ for the prior probabilities ($R_A$ stands for all classes regularization) and approximate $\overline{h}_c$ using mini-batches. The second regularization is needed to concentrate the probability distribution of each soft pseudo-label on a single class, thus avoiding the local optima in which the network might get stuck due to a weak guidance:

$$R_H = -\frac{1}{N} \sum_{i=1}^{N} \sum_{c=1}^{C} h_\theta^c(x_i) \log\left(h_\theta^c(x_i)\right), \tag{3}$$

where $h_\theta^c(x_i)$ denotes the $c$ class value of the softmax output $h_\theta(x_i)$ and again using mini-batches (i.e. $N$ is replaced by the mini-batch size) to approximate this term. This second regularization is the average per-sample entropy ($R_H$ stands for entropy regularization), a well-known regularization in SSL (Grandvalet & Bengio, 2004). Finally, the total semi-supervised loss is:

$$\ell = \ell^* + \lambda_A R_A + \lambda_H R_H, \tag{4}$$

where $\lambda_A$ and $\lambda_H$ control the contribution of each regularization term. We stress that this pseudo-labeling approach adapted from Tanaka et al. (2018) is far from the state-of-the-art for SSL (see Subsection 4.2), and are the mechanisms proposed in Subsection 3.1 which make pseudo-labeling a suitable alternative.

## 3.1 CONFIRMATION BIAS

Network predictions are, of course, sometimes incorrect. This situation is reinforced when incorrect predictions are used as labels for unlabeled samples, as it is the case in pseudo-labeling. Overfitting to incorrect pseudo-labels predicted by the network is known as confirmation bias. It is natural to think that reducing the confidence of the network on its predictions might alleviate this problem and improve generalization. Recently, mixup data augmentation (Zhang et al., 2018) introduced a strong regularization technique that combines data augmentation with label smoothing, which makes it potentially useful to deal with this bias. Mixup trains on convex combinations of sample pairs ($x_p$ and $x_q$) and corresponding labels ($y_p$ and $y_q$):

$$x = \delta x_p + (1 - \delta)x_q, \tag{5}$$

$$y = \delta y_p + (1 - \delta)y_q, \tag{6}$$

where $\delta \in \{0, 1\}$ is randomly sampled from a beta distribution $\mathcal{B}e(\alpha, \beta)$, with $\alpha = \beta$ (e.g. $\alpha = 1$ uniformly selects $\delta$). This combination regularizes the network to favor linear behavior in-between training samples, reducing oscillations in regions far from them. Additionally, Eq. 6 can be re-interpreted in the loss as $\ell^* = \delta \ell_p^* + (1 - \delta)\ell_q^*$, thus re-defining the loss $\ell^*$ used in Eq. 4 as:

$$\ell^* = -\sum_{i=1}^{N} \delta \left[\tilde{y}_{i,p}^T \log\left(h_\theta(x_i)\right)\right] + (1 - \delta)\left[\tilde{y}_{i,q}^T \log\left(h_\theta(x_i)\right)\right]. \tag{7}$$

As shown in (Thulasidasan et al., 2019), overconfidence in deep neural networks is a consequence of training on hard labels and it is the label smoothing effect from randomly combining $y_p$ and $y_q$ during mixup training that reduces prediction confidence and improves model calibration. In the semi-supervised context with pseudo-labeling, using soft-labels and mixup reduces overfitting to model predictions, which is especially important for unlabeled samples whose predictions are used as soft-labels. Note that training with mixup generates softmax outputs $h_\theta(x)$ for mixed inputs $x$, thus requiring a second forward pass with the original images to compute unmixed predictions. Mixup data augmentation alone might not effectively deal with confirmation bias when few labeled examples are provided. For example, when training with 500 labeled samples in CIFAR-10 and mini-batch size of 100, 1 clean sample per batch is seen, which is especially problematic at early stages of training where little correct guidance is provided. We find that setting a minimum number of labeled samples per mini-batch (as done in other works (Tarvainen & Valpola, 2017; Chen et al., 2018; Berthelot et al., 2019; Iscen et al., 2019)) provides a constant reinforcement with correct labels during training, reducing confirmation bias and helping to produce better pseudo-labels. Subsections 4.2 and 4.3 experimentally show that mixup, a minimum number of samples per mini-batch, and other techniques (dropout and data augmentation) reduce confirmation bias and make pseudo-labeling an effective alternative to consistency regularization.

## 4 EXPERIMENTAL WORK

### 4.1 DATASETS AND TRAINING

We use four image classification datasets, CIFAR-10/100 (Krizhevsky & Hinton, 2009), SVHN (Netzer et al., 2011) and Mini-ImageNet (Vinyals et al., 2016), to validate our approach. Part of the training images are labeled and the remaining are unlabeled. Following (Oliver et al., 2018), we use a validation set of 5K samples for CIFAR-10/100 for studying hyperparameters in Subsections 4.2 and 4.3. However, as done in (Athiwaratkun et al., 2019), we add the 5K samples back to the training set for comparisons in Subsection 4.4, where we report test results (model from the best epoch).

**CIFAR-10, CIFAR-100, and SVHN**   These datasets contain 10, 100, and 10 classes respectively, with 50K color images for training and 10K for testing in CIFAR-10/100 and 73257 images for training and 26032 for testing in SVHN. The three datasets have resolution 32×32. We perform experiments with a number of labeled images $N_l = 0.25$K, 0.5K, and 1K for SVHN and $N_l = 0.25$K, 0.5K, 1K, and 4K (4K and 10K) for CIFAR-10 (CIFAR-100). We use the well-known "13-CNN" architecture (Athiwaratkun et al., 2019) for CIFAR-10/100 and SVHN. We also experiment with a Wide ResNet-28-2 (WR-28) (Oliver et al., 2018) and a PreAct ResNet-18 (PR-18) (Zhang et al., 2018) in Subsection 4.3 to study the generalization to different architectures.

**Mini-ImageNet**   We emulate the semi-supervised learning setup Mini-ImageNet (Vinyals et al., 2016) (a subset of the well-known ImageNet (Deng et al., 2009) dataset) used in (Iscen et al., 2019). Train and test sets of 100 classes and 600 color images per class with resolution $84 \times 84$ are selected from ImageNet, as in (Ravi & Larochelle, 2017). 500 (100) images per-class are kept for train (test) splits. The train and test sets therefore contain 50k and 10k images. As with CIFAR-100, we experiment with a number of labeled images $N_l = 4$K and 10K. Following (Iscen et al., 2019), we use a ResNet-18 (RN-18) architecture (He et al., 2016a).

**Hyperparameters**   We use the typical configuration for CIFAR-10/100 and SVHN (Laine & Aila, 2017), and the same for Mini-ImageNet. Image normalization using dataset mean and standard deviation and subsequent data augmentation (Laine & Aila, 2017) by random horizontal flips and 2 (6) pixel translations for CIFAR and SVHN (Mini-ImageNet). Additionally, color jitter is applied as in (Asano et al., 2019) in Subsections 4.3 and 4.4 for higher robustness against confirmation bias. We train using SGD with momentum of 0.9, weight decay of $10^{-4}$, and batch size of 100. Training always starts with a high learning rate (0.1 in CIFAR and SVHN, and 0.2 in Mini-ImageNet), dividing it by ten twice during training. We train for CIFAR and Mini-ImageNet 400 epochs (reducing learning rate in epochs 250 and 350) and use 10 epoch warm-up with labeled data, while for SVHN we train 150 epochs (reducing learning rate in epochs 50 and 100) and use a longer warm-up of 150 epochs to start the pseudo-labeling with good predictions and leading to reliable convergence. We do not attempt careful tuning of the regularization weights $\lambda_A$ and $\lambda_H$ and just set them to 0.8 and 0.4 as done in (Tanaka et al., 2018) (see Appendix A.2 for an ablation study of these parameters). When using dropout, it is introduced between consecutive convolutional layers of ResNet blocks in WR-28, PR-18, and RN-18, while for 13-CNN we introduce it as in Laine & Aila (2017). Following (Athiwaratkun et al., 2019)[1], we use weight normalization (Salimans & Kingma, 2016) in all networks.

### 4.2 EFFECT OF MIXUP ON CONFIRMATION BIAS

This section demonstrates that carefully regularized pseudo-labeling is a suitable alternative for SSL. Figure 1 illustrates our approach on the "two moons" toy data. Figure 1 (left) shows the limitations of a naive pseudo-labeling adapted from (Tanaka et al., 2018), which fails to adapt to the structure in the unlabelled examples and results in a linear decision boundary. Figure 1 (middle) shows the effect of mixup, which alleviates confirmation bias to better model the structure and gives a smoother boundary. Figure 1 (right) shows that combining mixup with a minimum number of labeled samples $k$ per mini-batch improves the semi-supervised decision boundary.

Naive pseudo-labeling leads to overfitting the network predictions and high training accuracy in CIFAR-10/100. Table 1 (left) reports mixup effect in terms of validation error. Naive pseudo-labeling

---
[1]https://github.com/benathi/fastswa-semi-sup

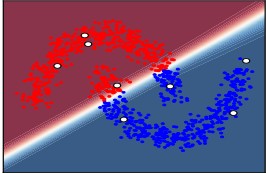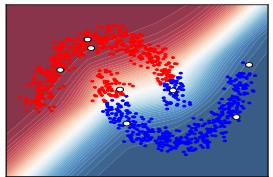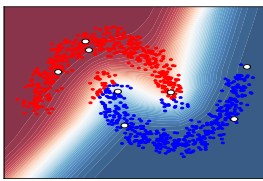

Figure 1: Pseudo-labeling in the "two moons" data (4 labels/class) for 1000 samples. From left to right: no mixup, mixup, and mixup with a minimum number of labeled samples per mini-batch. We use an NN classifier with one hidden layer with 50 hidden units as in (Miyato et al., 2018).

Table 1: Confirmation bias alleviation using mixup and a minimum number of $k$ labeled samples per mini-batch. Left: Validation error for naive pseudo-labeling without mixup (C), mixup (M), and alternatives with minimum $k$. Right: Study of the effect of $k$ on the validation error.

| | CIFAR-10 | | CIFAR-100 | | CIFAR-10 | | CIFAR-100 |
|---|---|---|---|---|---|---|---|
| Labeled images | 500 | 4000 | 4000 | Labeled images | 500 | 4000 | 4000 |
| C | 52.44 | 11.40 | 48.54 | $k = 8$ | **13.14** | 7.18 | 42.32 |
| C* ($k = 16$) | 35.08 | 10.90 | 46.60 | $k = 16$ | 13.68 | **6.90** | **38.78** |
| M | 32.10 | 7.16 | 41.80 | $k = 32$ | 14.58 | 7.06 | 39.62 |
| M *($k = 16$) | **13.68** | **6.90** | **38.78** | $k = 64$ | 19.40 | 8.20 | 46.28 |

leads to an error of 11.40/48.54 for CIFAR-10/100 when training with cross-entropy (C) loss for 4000 labels. This error can be greatly reduced when using mixup (M) to 7.16/41.80. However, when further reducing the number of labels to 500 in CIFAR-10, M is insufficient to ensure low-error (32.10). We propose to set a minimum number of samples $k$ per mini-batch to tackle the problem. Table 1 (right) studies this parameter $k$ when combined with mixup, showing that 16 samples per mini-batch works well for both CIFAR-10 and CIFAR-100, dramatically reducing error in all cases (e.g. in CIFAR-10 for 500 labels error is reduced from 32.10 to 13.68).

Confirmation bias causes a dramatic increase in the certainty of incorrect predictions during training. To demonstrate this behavior we compute the average cross-entropy of the softmax output with a uniform $\mathcal{U}$ across the classes in every epoch $t$ for all incorrectly predicted samples $\{x_{m_t}\}_{m_t=1}^{M_t}$ as: $r_t = -\frac{1}{M_t} \sum_{m_t=1}^{M_t} \mathcal{U}^T \log(h_\theta(x_{m_t}))$. Figure 2 shows that mixup and minimum $k$ are effective regularizers for reducing $r_t$, i.e. confirmation bias is reduced. We also experimented with using label noise regularizations (Xie et al., 2016), but setting a minimum $k$ proved more effective.

### 4.3 GENERALIZATION TO DIFFERENT ARCHITECTURES

There are examples in the recent literature (Kolesnikov et al., 2019) where moving from one architecture to another modifies the belief of which methods have a higher potential. Kolesnikov et al. (2019) show that skip-connections in ResNet architectures play a key role on the quality of learned representations, while most approaches in previous literature were systematically evaluated using AlexNet (A. Krizhevsky, 2012). Ulyanov et al. (2018) showed that different architectures lead different and useful image priors, highlighting the importance of exploring different networks. We, therefore, test our method with two more architectures: a Wide ResNet-28-2 (WR-28) (S. Zagoruyko, 2016) typically used in SSL (Oliver et al., 2018) (1.5M parameters) and a PreAct ResNet-18 (PR-18) (He et al., 2016b) used in the context of label noise (Zhang et al., 2018) (11M parameters). Table 2 presents the results for the 13-CNN (AlexNet-type) and these network architectures (ResNet-type). Our pseudo-labeling with mixup and $k = 16$ (M*) works well for 4000 and 500 labels across architectures, except for 500 labels for WR-28 where there is large error increase (29.50). This is due to a stronger confirmation bias in which labeled samples are not properly learned, while incorrect pseudo-labels are fit. Interestingly, PR-18 (11M) is more robust to confirmation bias than WR-28 (1.5M), while the 13-layer network (3M) has fewer parameters than PR-18 and achieves better performance. This suggests that the network architecture plays an important role, being a relevant prior for SSL with few labels.

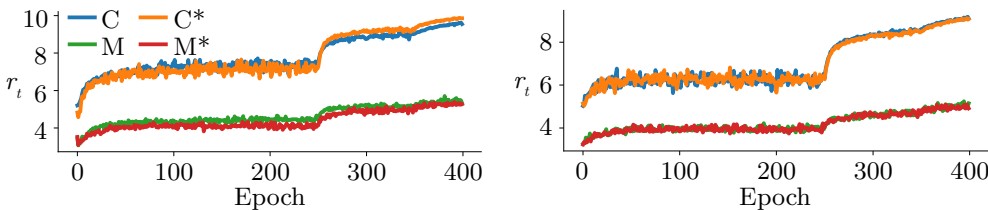

Figure 2: Example of certainty of incorrect predictions $r_t$ during training when using 500 (left) and 4000 (right) labeled images in CIFAR-10. Moving from cross-entropy (C) to mixup (M) reduces $r_t$, whereas adding a minimum number of samples per mini-batch (*) also helps in 500 labels, where M* (with slightly lower $r_t$ than M) is the only configuration that converges, as shown in Table 1 (left).

Table 2: Validation error across architectures is stabilized using dropout $p$ and data augmentation (A).

|  | M* | | M* ($p = 0.1$) | | M* ($p = 0.3$) | | M* ($p = 0.1$, A) | |
|---|---|---|---|---|---|---|---|---|
| Labeled images | 500 | 4000 | 500 | 4000 | 500 | 4000 | 500 | 4000 |
| 13-layer | 13.68 | 6.90 | 12.62 | 6.58 | 11.94 | 6.66 | **9.16** | **6.22** |
| WR-28 | 29.50 | **6.40** | 14.14 | 7.06 | 30.56 | 11.44 | **10.94** | 6.74 |
| PR-18 | **13.90** | **5.94** | 14.78 | 5.90 | 14.78 | 6.62 | 14.96 | 6.32 |

We found that dropout (Srivastava et al., 2014) and data augmentation is needed for good performance across all architectures. Table 2 shows that dropout $p = 0.1, 0.3$ helps in achieving better convergence in CIFAR-10, whereas adding color jitter as additional data augmentation (details in Subsection 4.1) further contributes to error reduction. Note that the quality of pseudo-labels is key, so it is essential to disable dropout to prevent corruption when computing these in the second forward pass. We similarly disable data augmentation in the second forward pass, which consistently improves performance. This configuration is used for comparison with the state-of-the-art in Subsection 4.4.

### 4.4 COMPARISON WITH THE STATE-OF-THE-ART

We compare our pseudo-labeling approach against related work that makes use of the 13-CNN (Tarvainen & Valpola, 2017) in CIFAR-10/100: $\Pi$ model (Laine & Aila, 2017), TE (Laine & Aila, 2017), MT (Tarvainen & Valpola, 2017), $\Pi$ model-SN (Luo et al., 2018), MA-DNN (Chen et al., 2018), Deep-Co (Qiao et al., 2018), TSSDL (Shi et al., 2018), LP (Iscen et al., 2019), CCL (Li et al., 2019), fast-SWA (Athiwaratkun et al., 2019) and ICT (Verma et al., 2019). Table 3 divides methods into those based on consistency regularization and pseudo-labeling. Note that we include pseudo-labeling approaches combined with consistency regularization ones (e.g. MT) in the consistency regularization set. The proposed approach clearly outperforms consistency regularization methods, as well as other purely pseudo-labeling approaches and their combination with consistency regularization methods in CIFAR-10/100 and SVHN (250 labels are reported in Table 3 and extended results are in the Appendix A.3). These results demonstrate the generalization of the proposed approach compared to other methods that fail when decreasing the number of labels. Furthermore, Table 4 (left) demonstrates that the proposed approach successfully scales to higher resolution images, obtaining an over 10 point margin on the best related work in Mini-ImageNet. Note that all supervised baselines are reported using the same data augmentation and dropout as in the proposed pseudo-labeling.

Table 4 (right) compares against recent consistency regularization approaches that use mixup. We achieve better performance than ICT (Verma et al., 2019), while being competitive with MM (Berthelot et al., 2019) for 500 and 4000 labels using WR-28. Regarding PR-18, we converge to reasonable performance for 4000 and 500 labels, whereas for 250 we do not. Finally, the 13-CNN robustly converges even for 250 labels where we obtain 9.37 test error (see Appendix A.1 for some details on different architectures convergence). Therefore, these results suggest that it is worth exploring the relationship between number of labels, dataset complexity and architecture type. As shown in Subsection 4.3, dropout and additional data augmentation help with 500 labels/class across architectures, but are insufficient for 250 labels. Better data augmentation (Ho et al., 2019) or self-supervised pre-training (Rebuffi et al., 2019) might overcome this challenge. Furthermore,

Table 3: Test error in CIFAR-10/100 and SVHN for the proposed approach using the 13-CNN network. (*) denotes that we have run the algorithm. Bold indicates lowest error. We report average and standard deviation of 3 runs with different labeled/unlabeled splits.

| | CIFAR-10 | | | CIFAR-100 | | SVHN |
|---|---|---|---|---|---|---|
| Labeled images | 500 | 1000 | 4000 | 4000 | 10000 | 250 |
| Supervised (C)* | $43.64 \pm 1.21$ | $34.83 \pm 1.15$ | $19.26 \pm 0.26$ | $54.49 \pm 0.53$ | $41.14 \pm 0.26$ | - |
| Supervised (M)* | $37.60 \pm 0.65$ | $28.59 \pm 1.21$ | $15.94 \pm 0.26$ | $52.70 \pm 0.28$ | $39.42 \pm 0.37$ | - |
| Consistency regularization methods | | | | | | |
| $\Pi$ model | - | - | $12.36 \pm 0.31$ | - | $39.19 \pm 0.36$ | $9.69 \pm 0.92$ |
| TE | - | - | $12.16 \pm 0.24$ | - | $38.65 \pm 0.51$ | - |
| MT | $27.45 \pm 2.64$ | $19.04 \pm 0.51$ | $11.41 \pm 0.25$ | $45.36 \pm 0.49$ | $36.08 \pm 0.51$ | $4.35 \pm 0.50$ |
| $\Pi$ model-SN | - | $21.23 \pm 1.27$ | $11.00 \pm 0.13$ | - | $37.97 \pm 0.29$ | $5.07 \pm 0.25$ |
| MA-DNN | - | - | $11.91 \pm 0.22$ | - | $34.51 \pm 0.61$ | - |
| Deep-Co | - | - | $9.03 \pm 0.18$ | - | $38.77 \pm 0.28$ | - |
| MT-TSSDL | - | $18.41 \pm 0.92$ | $9.30 \pm 0.55$ | - | - | $4.09 \pm 0.42$ |
| MT-LP | $24.02 \pm 2.44$ | $16.93 \pm 0.70$ | $10.61 \pm 0.28$ | $43.73 \pm 0.20$ | $35.92 \pm 0.47$ | - |
| MT-CCL | - | $16.99 \pm 0.71$ | $10.63 \pm 0.22$ | - | $34.81 \pm 0.52$ | - |
| MT-fast-SWA | - | $15.58 \pm 0.12$ | $9.05 \pm 0.21$ | - | $34.10 \pm 0.31$ | - |
| ICT | - | $15.48 \pm 0.78$ | $7.29 \pm 0.02$ | - | - | $4.78 \pm 0.68$ |
| Pseudo-labeling methods | | | | | | |
| TSSDL | - | $21.13 \pm 1.17$ | $10.90 \pm 0.23$ | - | - | $5.02 \pm 0.26$ |
| LP | $32.40 \pm 1.80$ | $22.02 \pm 0.88$ | $12.69 \pm 0.29$ | $46.20 \pm 0.76$ | $38.43 \pm 1.88$ | - |
| Ours* | $\mathbf{8.80 \pm 0.45}$ | $\mathbf{6.85 \pm 0.15}$ | $\mathbf{5.97 \pm 0.15}$ | $\mathbf{37.55 \pm 1.09}$ | $\mathbf{32.15 \pm 0.50}$ | $\mathbf{3.66 \pm 0.12}$ |

Table 4: Test error in Mini-ImageNet (left) and CIFAR-10 with few labeled samples (right). (*) denotes that we have run the algorithm. Bold indicates lowest error. We report average and standard deviation of 3 runs with different labeled/unlabeled splits.

| Labeled images | 4000 | 10000 |
|---|---|---|
| Supervised (C)* | $75.69 \pm 0.24$ | $63.24 \pm 0.33$ |
| Supervised (M)* | $72.03 \pm 0.21$ | $59.96 \pm 0.40$ |
| Consistency regularization methods | | |
| MT | $72.51 \pm 0.22$ | $57.55 \pm 1.11$ |
| MT-LP | $72.78 \pm 0.15$ | $57.35 \pm 1.66$ |
| Pseudo-labeling methods | | |
| LP | $70.29 \pm 0.81$ | $57.58 \pm 1.47$ |
| Ours* | $\mathbf{56.49 \pm 0.51}$ | $\mathbf{46.08 \pm 0.11}$ |

| Labeled images | 250 | 500 | 4000 |
|---|---|---|---|
| MM (WR-28) | $\mathbf{11.08 \pm 0.87}$ | $\mathbf{9.65 \pm 0.94}$ | $\mathbf{6.24 \pm 0.06}$ |
| ICT* (WR-28) | $52.19 \pm 1.54$ | $42.33 \pm 0.08$ | $7.26 \pm 0.04$ |
| Ours* (WR-28) | $24.81 \pm 5.35$ | $14.25 \pm 0.86$ | $6.28 \pm 0.3$ |
| Ours* (13-CNN) | $\mathbf{9.37 \pm 0.12}$ | $\mathbf{8.80 \pm 0.45}$ | $5.97 \pm 0.15$ |
| Ours* (PR-18) | $23.86 \pm 4.82$ | $12.16 \pm 1.06$ | $\mathbf{5.86 \pm 0.17}$ |

hyperparameters such as the regularization weights $\lambda_A = 0.8$ and $\lambda_H = 0.4$ from Eq. 4 and the mixup $\alpha$ require further study. However, it is already interesting that a straightforward modification of pseudo-labeling, designed to tackle confirmation bias, gives a competitive semi-supervised learning approach, without any consistency regularization, and future work should take this into account.

## 5 CONCLUSIONS

This paper presented a semi-supervised learning approach for image classification based on pseudo-labeling. We proposed to directly use the network predictions as soft pseudo-labels for unlabeled data together with mixup augmentation, a minimum number of labeled samples per mini-batch, dropout and data augmentation to alleviate confirmation bias. This conceptually simple approach outperforms related work in four datasets, demonstrating that pseudo-labeling is a suitable alternative to the dominant approach in recent literature: consistency-regularization. The proposed approach is, to the best of our knowledge, both simpler and more accurate than most recent approaches. Future work should explore SSL in class-unbalanced and large-scale datasets, synergies of pseudo-labelling and consistency regularization, and careful hyperparameter tuning.

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

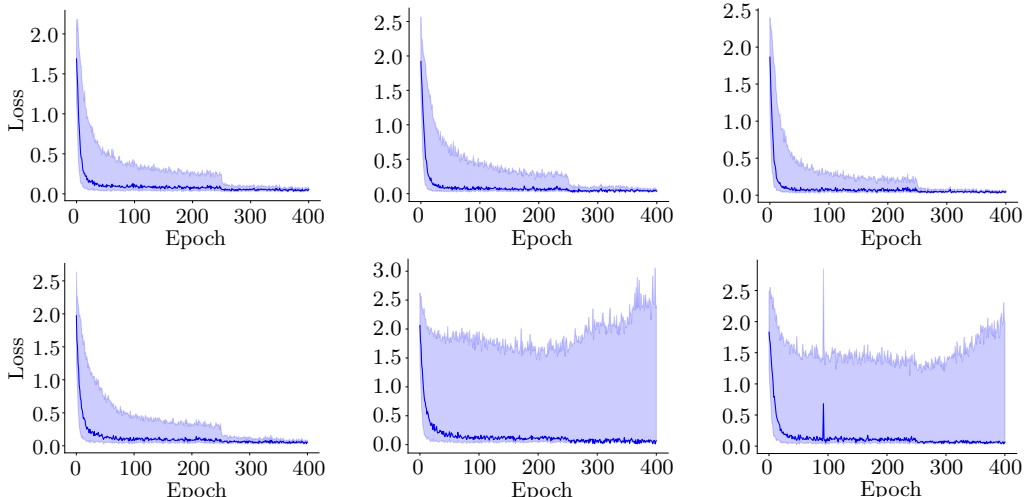

Figure 3: Cross-entropy loss for labeled samples. First (second) row show 500 (250) labels in CIFAR-10. From left to right: 13-CNN, WR-28 and PR-18. The heavy lines represent the median losses and the shaded areas are the interquartile ranges.

## A APPENDIX

### A.1 CONVERGENCE FOR FEW LABELS

Figure 3 presents the cross-entropy loss for labeled samples when training with 13-CNN, WR-28 and PR-18 and using 500 and 250 labels in CIFAR-10. This loss is a good indicator of a robust convergence to reasonable performance as the interquartile range for cases failing (250 labels for WR-28 and PR-18) is much higher.

### A.2 EXTENDED HYPERPARAMETERS STUDY

This subsection studies the effect of $\alpha$, $\lambda_A$, and $\lambda_H$ hyperparameters of our pseudo-labeling approach. Tables 5 and 6 report the validation error in CIFAR-10 using 500 and 4000 labels for, respectively, $\alpha$ and $\lambda_A$ and $\lambda_H$ . Note that we keep the same configuration used in Subsection 4.2 with $k = 16$, i.e. no dropout or additional data augmentation is used. Table 5 results suggest that $\alpha = 4$ and $\alpha = 8$ values might further improve the reported results using $\alpha = 1$. However, we experimented on CIFAR-10 with 500 labels using the final configuration (adding dropout and additional data augmentation) and observed marginal differences (8.54 with $\alpha = 4$, which is within the error range of the $8.80 \pm 0.45$ obtained with $\alpha = 1$), thus suggesting that stronger mixup regularization might not be additive to dropout and extra data augmentation in our case. Table 6 shows that our configuration ($\lambda_A = 0.8$ and $\lambda_H = 0.4$) adopted from (Tanaka et al., 2018) is very close to the best performance in this experiment where marginal improvements are achieved. In conclusion, more careful hyperparameter tuning might slighltly improve reported results in the paper, while the configuration selected is already good and generalizes across datasets.

### A.3 SVHN EXPPERIMENTS

Table 7 reports a comparison of different state-of-the-art algorithms in SVHN using the 13-CNN network. We train 150 epochs using labeled samples (i.e. warm-up) to compute initial pseudo-labels. Then, we train 150 epochs reducing the learning rate in epochs 50 and 100. The long warm-up makes the method robust to lower levels of labeled samples in SVHN. We also experimented in CIFAR-10 with longer warm-up (our results are reported using 10 epochs) and found that results are in the same error range already reported.

Table 5: Validation error for different values of the $\alpha$ parameter from Mixup. Bold indicates lowest error. Underlined values indicate the results of the configuration used.

| Labeled images: | 500 | | | | 4000 | | | |
|---|---|---|---|---|---|---|---|---|
| $\alpha$ | 0.1 | 1 | 4 | 8 | 0.1 | 1 | 4 | 8 |
| | 23.18 | 13.68 | **10.60** | 11.04 | 8.58 | 6.90 | **6.56** | 6.68 |

Table 6: Validation error for different values of $\lambda_A$ and $\lambda_H$. Bold indicates lowest error. Underlined values indicate the results of the configuration used.

| Labeled images: | 500 | | | | 4000 | | | |
|---|---|---|---|---|---|---|---|---|
| $\lambda_A/\lambda_H$ | 0.1 | 0.4 | 0.8 | 2 | 0.1 | 0.4 | 0.8 | 2 |
| 0.1 | 22.94 | 29.64 | 60.76 | 83.96 | 7.22 | **6.88** | 7.74 | 33.98 |
| 0.4 | 20.92 | **12.88** | 17.62 | 38.40 | 7.18 | 6.96 | 7.18 | 8.82 |
| 0.8 | 23.50 | 13.68 | 14.72 | 25.92 | 7.24 | 6.90 | 7.18 | 8.78 |
| 2 | 31.30 | 14.80 | 14.62 | 23.40 | 8.16 | 7.28 | 7.40 | 8.64 |

Table 7: Test error in SVHN for the proposed approach using the 13-CNN network. (*) denotes that we have run the algorithm. Bold indicates lowest error. We report average and standard deviation of 3 runs with different labeled/unlabeled splits.

| Labeled images | 250 | 500 | 1000 |
|---|---|---|---|
| Supervised (C)* | 43.60±3.35 | 22.67±2.80 | 13.32±0.89 |
| Supervised (M)* | 53.15±6.54 | 20.74±0.80 | 11.66±0.17 |
| Consistency regularization methods | | | |
| $\Pi$ model | $9.69 \pm 0.92$ | $6.83 \pm 0.66$ | $4.95 \pm 0.26$ |
| TE | - | $5.12 \pm 0.13$ | $4.42 \pm 0.16$ |
| MT | $4.35 \pm 0.50$ | $4.18 \pm 0.27$ | $3.95 \pm 0.19$ |
| $\Pi$ model-SN | $5.07 \pm 0.25$ | $4.52 \pm 0.30$ | $3.82 \pm 0.25$ |
| MA-DNN | - | - | $4.21 \pm 0.12$ |
| Deep-Co | - | - | $3.61 \pm 0.15$ |
| MT-TSSDL | $4.09 \pm 0.42$ | $3.90 \pm 0.27$ | $\mathbf{3.35 \pm 0.27}$ |
| ICT | $4.78 \pm 0.68$ | $4.23 \pm 0.15$ | $3.89 \pm 0.04$ |
| Pseudo-labeling methods | | | |
| TSSDL | $5.02 \pm 0.26$ | $4.32 \pm 0.30$ | $3.80 \pm 0.27$ |
| Ours* | $\mathbf{3.66 \pm 0.12}$ | $\mathbf{3.64 \pm 0.04}$ | $3.55 \pm 0.08$ |

