# OpenReview forum: "Pseudo-Labeling and Confirmation Bias in Deep Semi-Supervised Learning"
_ICLR.cc/2020/Conference — Reject_

### Official Review · AnonReviewer1 · 2019-10-08
**Official Blind Review #1**

**Rating:** 8

**Review:**

OVERALL:
I think this paper is worth accepting.
All modern semi-supervised learning techniques use consistency regularization somehow,
and this paper shows that you can get away with just using pseudo-labeling combined with some
engineering to route around the main issue with pseudo labeling (which is apparently called confirmation bias,
though I hadn't heard that, and I don't like it as a name because it's confusing).

Neither MixUp nor the idea of fixing some number of labeled elements in a minibatch is new,
but that's not the point - we thought one thing, and this paper suggests that
we were wrong about that thing - to me this is exactly the sort of paper it's good to have at conferences.

I would change the framing slightly.
You're not showing that pseudo-labeling can be useful, because many techniques already incorporate a form of pseudo-labeling.
Instead, you're showing you can get away without consistency regularization.

A potential improvement:
If you add up this techique with some of the most recent SLL techniques based on consistency regularization somehow,
does it do better, or are they both acting via the same mechanism?

DETAILED COMMENTS:
> , contrary to previous evidences on pseudo-labeling capabilities (Oliver et al., 2018),
It's not really contrary to the findings of that paper, since you've totally changed the
technique compared to what's evaluated in that paper.

> n (Berthelo et al., 2019)
It's Berthelot

> and are the mechanisms proposed in Subsection 3.1
Doesn't quite parse

> Network predictions are, of course, sometimes incorrect.
This is a great line.

> We use three image classification datasets...
Why not use SVHN, which is by now super standard for SSL papers?

> , we add the 5K samples back to the training set for comparison
with the state-of-the-art in Subsection 4.4,
This is *allowed* from the perspective of reporting a valid test accuracy,
but if other papers don't do that, it kind of mucks up the comparison, no?

Fig 1 is nice, but why does the effect not seem to be symmetric about the
blue and the red blobs?

> architecture plays and important role


> However, it is already interesting that...  and that future work should take this into account.
This sentence doesn't quite make sense

Re table 4:
I'm curious how e.g. MixMatch would fare w/ the 13-CNN network.
I am surprised that the change from WRN -> 13-CNN matters so much.


**Experience Assessment:**

I have published one or two papers in this area.

**Review Assessment: Checking Correctness Of Derivations And Theory:**

N/A

**Review Assessment: Checking Correctness Of Experiments:**

I assessed the sensibility of the experiments.

**Review Assessment: Thoroughness In Paper Reading:**

I read the paper at least twice and used my best judgement in assessing the paper.

---

> ### Author Response · Authors · 2019-11-13
> **RE: Thank you for the review**
>
> Thank you for your review and useful feedback. We have corrected the minor typos reported in the updated version of the manuscript.

---

> ### Author Response · Authors · 2019-11-13
> **RE: Full response**
>
>
> - Regarding “confirmation bias” term
>
> We adopted this term from other papers: Tarvainen & Valpola, 2019 (MT) and Li et al., 2019 (CCL), and note that is also named the noise accumulation problem (Zhang et al., 2016). In psychology it is defined as “the tendency to search for, interpret, favor, and recall information in a way that affirms one's prior beliefs or hypotheses” [1]. In the context of Deep Neural Networks the term can be explained as: “the model is prone to confirm the previous predictions and resist new changes” (CCL). Dealing with confirmation bias has been studied in MT and CCL, where they report a behaviour like the one we encounter, but for consistency regularization approaches. The issue they find is that when increasing the weight of the consistency regularization term, it outweighs the cross-entropy term and prevents the learning of new information (see Figure 1 in MT).
>
> [1] Plous, Scott, 1993, The Psychology of Judgment and Decision Making, p. 233.
>
>
> - Regarding a slight change in the framing
>
> We have changed the framing slightly following your suggestion to reflect that we show that pseudo-labeling does not need consistency regularization and prevent possible misunderstandings on previous capabilities already shown by pseudo-labeling when combined with consistency regularization. The following changes were made to the manuscript.
>
> Abstract: “These results demonstrate that pseudo-labeling can outperform consistency regularization methods, while the opposite was supposed in previous work.”
>
> Introduction: “This paper explores pseudo-labeling for semi-supervised deep learning from the network predictions and shows that, contrary to previous attempts on pseudo-labeling (Iscen et al., 2019, Oliver et al., 2018, Shi et al., 2018), simple modifications to prevent confirmation bias lead to state-of-the-art performance without adding consistency regularization strategies.”
>
>
> - Regarding combination of our approach with consistency regularization
>
> We agree in that consistency regularization might further improve our approach as previous evidence shows that pseudo-labeling and consistency regularization encounter benefits when combined (Iscen et al., 2019, Shi et al., 2018). Since pseudo-labeling and consistency regularization represent different forms of leveraging unlabeled data, they might encounter some beneficial complementarity. However, as we have added to the conclusions section, we leave this for future work as want to stress the potential of pseudo-labeling by itself.
>
> - Regarding SVHN
>
> We have updated the paper to include results on SVHN dataset in Table 3 and in Appendix A.3 using the 13-CNN network. We have experimented with 250, 500, and 1000 labeled examples and obtained, respectively,  3.66 ± 0.12, 3.64 ± 0.04, and 3.55 ± 0.08 (these are state-of-the-art results on-par with top-performing consistency regularization approaches). It is important to highlight that to assure convergence to reasonable performance with few labels we had to perform a longer warm-up period (150 epochs) to improve the quality of pseudo-labels in early training epochs (the same modification in CIFAR-10 using 250 labeled examples achieves a similar performance inside the range of error reported in the paper).
>
> - Regarding the use of the validation set
>
> We decided to adopt the criterion of separating a small validation subset from the training data and then replacing it due to the same approach used in the 2019 ICLR paper by Athiwaratkun et al.. This ensures that 10K samples of the test subset are never seen during hyperparameter tuning and 50K samples are used for training. All numbers reported in the tables were obtained under the same conditions as were used in our experiments, i.e. using 50K training examples and 10K test examples. The only exception is ICT (Verma et al., 2019), where they use the labeled samples both with labels and without labels, thus slightly increasing the amount of unlabeled examples above 50K.
>
> - Regarding the toy examples asymmetry
>
> The asymmetry seen in Fig 1 is due to the fact that the samples that have labels do not form a symmetric pattern. These samples more strongly affect the location of the decision boundary than the unlabelled samples. This observation can be seen as well in figures reported in (Rebuffi et al., 2019) and (Verma et al., 2019).
>
>
> - Regarding MixMatch (MM) with different architectures
>
> We think that MM is a very powerful approach that would not have issues when run with the 13-CNN layer network. Also, as reported in (Kolesnikov et al. 2019), the network architecture may play a very important role, as shown for self-supervised learning with VGG-type and ResNet-type architectures. We observed something similar for semi-supervised learning with pseudo-labeling. Future work should take into consideration that trying multiple architectures might reveal interesting results.

---

### Official Review · AnonReviewer3 · 2019-10-20
**Official Blind Review #3**

**Rating:** 3

**Review:**

This paper proposes to combine pseudo-labelling with MixUp to tackle the semi-supervised classification problem. My problem is that "MixMatch: A Holistic Approach to Semi-Supervised Learning" by Berthelot et al. is very similar with just a few differences on the pseudo-labelling part. Could you stress more the difference between your paper and their paper ? Because I might be wrong about it.

Pros:
* Good results on C10
* A clear related work section that divides the existing works in pseudo labelling vs consistency
* Interesting results about the effects of using different architectures. I also like the ablation study.

Weaknesses:
* Usually, SVHN is also among the tested datasets
* The pseudo labelling part is a bit unclear.For example, do you just refresh the pseudo-labels at the end of each epoch ?
* minor: a typo with "and important role"

If there was not an existing paper already using MixUp, I would have leaned towards acceptance. You can still motivate the differences with the MixMatch paper.

**Experience Assessment:**

I have published one or two papers in this area.

**Review Assessment: Checking Correctness Of Derivations And Theory:**

N/A

**Review Assessment: Checking Correctness Of Experiments:**

I carefully checked the experiments.

**Review Assessment: Thoroughness In Paper Reading:**

N/A

---

> ### Author Response · Authors · 2019-11-13
> **RE: Full response**
>
> Thank you for your review and useful feedback. We have corrected the typo reported in the updated version of the manuscript.
>
> - Regarding the difference with MixMatch (MM)
>
> MM is a powerful consistency regularization approach. Here we focus on pseudo-labeling. This is a substantial difference because the type of guidance that these two approaches use is based on different ideas. To highlight this difference we have modified the corresponding paragraph in the introduction (new text in italics): “Recent approaches in image classification primarily focus on exploiting the consistency in the predictions for the same sample under different perturbations (consistency regularization) (Sajjadi et al., 2016; Li et al., 2019), while other approaches directly generate labels for the unlabeled data to guide the learning process (pseudo-labeling) (Lee, 2013; Iscen et al., 2019). These two alternatives differ importantly in the mechanism they use to exploit unlabeled samples.”
>
> Therefore, yes, both papers use mixup, but they differ importantly how unlabeled samples are used. Our method uses pseudo-labeling, which was thought not to work without combining it with consistency regularization, and we demonstrate that when dealing with confirmation bias (which we tackle mainly with mixup) it achieves state-of-the-art results. We think that modifying previous beliefs is an important contribution that we support with: a toy problem visualization in Figure 1, extensive analysis of different hyperparameters (adding and removing mixup in Table 1, the importance of setting a minimum number of samples per mini-batch in Table 1, dropout and data augmentation importance in Table 2 and newly added hyperparameter studies as suggested by Reviewer#2 in Appendix A.3), and extensive evaluations in CIFAR-10/100, Mini-ImageNet, and (newly added) SVHN (Table 3 in the paper and Table 7 in the Appendix A.3).
>
> - Regarding SVHN
>
> Following your suggestion, we evaluated our approach in the popular SVHN dataset obtaining state-of-the-art results. We use 250, 500, and 1000 labeled examples (uniformly distributed across classes as done in the related work), obtaining errors of 3.66 ± 0.12, 3.64 ± 0.04, and 3.55 ± 0.08 (these are state-of-the-art results on-par with top-performing consistency regularization approaches). We use the 13-CNN network and train 150 epochs (starting with learning rate 0.1 and dividing it by 10 twice in epochs 50 and 100). The modification needed to operate in this dataset was to perform a longer warm up stage to start the pseudo-labeling with good predictions and leading to reliable convergence (the same modification in CIFAR-10 using 250 labeled examples achieves a performance inside the range of error reported in the paper). We include SVHN results for 250 labels in Table 3, while complete results are provided in Appendix A.3.
>
> - Regarding pseudo-labels update
>
> Thank you for noting the confusion. We update the pseudo-labels at the end of every epoch. We have updated the text in between Eq.1 and 2 in Section 3 to read: “In particular, we store the softmax predictions h_θ(x_i) of the network in every mini-batch of an epoch and use them to modify the soft pseudo-label y ̃ for the N_u unlabeled samples at the end of every epoch”. We changed “at the end of the epoch” by “at the end of every epoch” to make it clear.

---

### Official Review · AnonReviewer2 · 2019-10-21
**Official Blind Review #2**

**Rating:** 3

**Review:**

Summary: This paper focuses on the semi-supervised learning problem, and proposes a way to improve previous pseudo-labeling methods.  In pseudo-labeling, there is an issue called confirmation bias, which accumulates the early errors of wrong pseudo labels.  By adding some simple tricks such as adding mixup augmentation and setting a minimum number of labeled samples per mini-batch, the confirmation bias is shown to be reduced, leading to an improvement in accuracy.  Experiments demonstrate that the additional tricks are meaningful and makes pseudo-labeling better than many baseline methods for semi-superivsed learning, including state-of-the-art consistency regularization methods.


Pros: This is an interesting paper with a clear motivation, which is to fix the so-called confirmation bias that appears in pseudo-labeling methods for semi-supervised learning.  Although the tricks introduced in the paper (mixup and changing the mini-batch selection rules) themselves are not novel, they make the proposed method simple.  It is also shown to be meaningful in reducing the confirmation bias in Table 1 and Figure 2, achieving the original goal of the paper.


Cons: The weakness of the paper is that the intuition or the motivation behind the design of the proposed method is not so clear.  Using mixup is justified by the reason that mixup gives better confidence calibration.  This is important for pseudo-labeling methods, because soft-label output predictions are used as pseudo labels.  On the other hand, however, it was not so obvious why a minimum number of labeled samples per mini-batch was considered.  Can we consider further extensions such as minimum number of labeled samples per mini-batch & per class?  (Perhaps the discussions about mixup and soft labels in the last paragraph of Section 3 should be more emphasized, for example in the last paragraph of the Introduction section.)

Related to the weakness above, it is hard to see how far the regularization effects of adding mixup and mini-batch sampling rules are contributing to add synergy to the pseudo-labeling methods.  This is partially answered with Figure 2, but it would make this easier to see if the experiments included stronger baselines, e.g., by adding the same regularization tricks to consistency regularization methods, perhaps in Table 3.

Finally, since future work on pseudo labels will follow this paper’s setup, hyperparameters such as lamba_A, lambda_H, and alpha should be chosen carefully instead of fixing them.


Other minor comments (that did not impact the score):

- In reference section, "Z. MaXiaoyu Tao" seems to combine two authors.

- Table 3 never appears in the text.  In Section 4.4, "The table" in the second sentence can be changed to "Table 3".

- "architecture plays and important role" --> "architecture plays an important role"

- In Table 2, "+" signs make it look like an equation.  I suggest using commas instead.

-  "ResNet arquitectures" --> "ResNet architectures"

~~~~~
Thank you for the response and for the additional discussions that were included in the updated paper.

**Experience Assessment:**

I have read many papers in this area.

**Review Assessment: Checking Correctness Of Derivations And Theory:**

N/A

**Review Assessment: Checking Correctness Of Experiments:**

I assessed the sensibility of the experiments.

**Review Assessment: Thoroughness In Paper Reading:**

I read the paper at least twice and used my best judgement in assessing the paper.

---

> ### Author Response · Authors · 2019-11-13
> **RE: Thank you for the review**
>
> Thank you for your review and useful feedback. We have corrected the minor typos reported in the updated version of the manuscript.

---

> ### Author Response · Authors · 2019-11-13
> **RE: Full response**
>
>
> -Regarding the motivation behind the design of our approach
>
> As you have pointed out: we address confirmation bias to make pseudo-labeling (without consistency regularization) a suitable approach for semi-supervised learning (SSL). There might be other solutions aside from those proposed in this work, but we find mixup augmentation to be very effective and the minimum number of samples per mini-batch to be key when reducing the labeled examples. It is true that these “tricks” are not new, but we believe that the main contribution of the paper is to demonstrate that a conceptually simple pseudo-labeling approach can achieve state-of-the-art results for SSL without being combined with consistency regularization, which is in opposition to previous beliefs.
>
> Mixup reduces the general confidence of the network (as shown in Thulasidasan et al. 2019) and this calibration effect directly tackles confirmation bias and helps pseudo-labeling on being a successful approach for SSL (as shown in Fig. 2).
>
> A minimum number of k samples per mini-batch is a common practice (MixMatch, MT, LP, MA-DNN) that is seldom reported formally. Nevertheless, Tab. 1 shows its importance when reducing the number of labeled samples and Figure 2 (left) shows that in these cases it further reduces confirmation bias. We agree with your observation that we have not sufficiently motivated the use of this parameter, thus we have extended the discussion at the end of Subsec. 3.1 as you suggested.
>
> Regarding your suggestion to study a minimum number of classes per batch and per class: we agree that this may be useful when having unbalanced data to prevent bias towards predicting certain classes. This is not the case in the datasets studied in this paper (note that the newly added SVHN is unbalanced, but the labeled set is balanced thus hiding the unbalanced nature of the dataset from a practical perspective). In cases where the number of classes increases, however, it becomes infeasible to ensure a minimum number of samples per class due to batch size restrictions (e.g. training a network with CIFAR100 and a batch of 100 samples allows only for a single sample per class and no unlabeled samples). We therefore think that your suggestion points an important issue that should be addressed in future work: are SSL approaches robust in unbalanced scenarios where the unbalanced nature is not known a-priori? We have pointed out this observation in Sec. 5.
>
> -Regarding stronger baselines when mixup (and other strategies) are combined with related work approaches
>
> We agree that strong baselines are important. The paper includes ICT (Verma et al., 2019) and MM (Berthelot et al., 2019) in Tab. 3 and 4 for this reason. Both ICT and MM are recent and top-performing consistency regularization approaches that use mixup data augmentation. Regarding a minimum number of samples per batch: MM, MT, and MA-DNN are consistency regularization methods that use it, while LP is a pseudo-labeling approach that also adopt it. We show that our method outperforms these methods (except the consistency regularization method MM for which we are almost on-par). This supports the claim that pseudo-labeling does not require consistency regularization to achieve state-of-the-art results. Furthermore, pseudo-labeling approaches (TSSDL and LP) have previously been shown to benefit from their combination with consistency regularization; we further show that cleaner pseudo-labels (without dropout or data augmentation) lead to better performance. We have added to Sec. 5 that future work should explore the synergies that consistency regularization and a strong pseudo-labeling method like the one proposed might result in.
>
> -Regarding the study of hyperparameters lambda_A, lambda_B and alpha
>
> Following your suggestion we have studied these hyperparameters and report the results in Tab. 5 and 6 of Appendix A.2 to show our method’s behavior with different values of these parameters (note that we were already studying key characteristics such as using mixup or not, the minimum number of labeled examples parameter k, and synergies of mixup and dropout). These experiments confirm that the configuration selected is close to the best results achievable by tuning the hyperparameters. Regarding lambda_A and lambda_B: we use the values suggested in (Tanaka et al., 2018) which are very close to the top performance observed in the new experiment in Tab. 7. We use alpha=1 for the mixup hyperparam. as done in [2]. This value means that the mixing coefficient delta in mixup is uniformly sampled between 0 and 1, enabling a wide variety of data augmentations. Tab. 5 shows that alpha=1 might be surpassed by other alphas (values 4 and 8). However, these improvements are marginal and do not affect the contribution that mixup helps pseudo-labeling in reducing confirmation bias and leads to good performance.
>
> [2] V. Verma et al., Manifold Mixup: Better Representations by Interpolating Hidden States. ICML 2019.

---

### Public Comment · ~Bao_Wang1 · 2019-10-19
**It is a very cool idea**

Hi, interpolation is a very cool idea in deep semi-supervised learning. Here I would like to point out a few papers that might be of interest to you.

1. B. Wang, et al. Deep Neural Nets with Interpolating Function as Output Activation, NeurIPS 2018.

2. B. Wang, et al. Adversarial Defense via Data Dependent Activation Function and Total Variation Minimization, arXiv:1809.08516 2018

3. B. Wang, et al. Graph Interpolating Activation Improves Both Natural and Robust Accuracies in Data-Efficient Deep Learning, arXiv:1907.06800 2019.

Thanks for your attention.

---

### Decision · Program_Chairs · 2019-12-19

**Decision:**

Reject

**Comment:**

The paper focuses on semi-supervised learning and presents a pseudo labeling-based approach with i) mixup (Zhang et al. 2018); ii) keeping $k$ labelled examples in each minibatch.

The paper is clear and well-written; it presents a simple and empirically effective idea. Reviewers appreciate the nice proof of concept on the two-moons dataset, the fact that the approach is validated with different architectures. Some details would need to be clarified, e.g. about the dropout control.

A main contribution of the paper is to show that pseudo-labelling plus the combination of mixup and certainty (keeping $k$ labelled examples in each minibatch) can outperform the state of the art based on consistency regularization methods, while being simpler and computationally much less demanding.

While the paper does a good job of showing that "it works", the reader however misses some discussion about "why it works". It is most interesting that the performances are not improving with $k$ (Table 1). An in-depth analysis of the trade-off between the uncertainty (through mix-up and the entropy of the pseudo-labels) and certainty, and how it impacts the performance, would be appreciated. You might consider monitoring how this trade-off evolves along learning; I suspect that evolving $k$ along the epochs might make sense;  the question is to find a simple way to control online this hyper-parameter.

The area chair encourages the authors to continue this very promising path of research, and dig a little bit deeper, considering the question of optimizing the trade-off between certainty and uncertainty along the training trajectory.